# Development of a Combined Exercise and Cognitive Stimulation Intervention for People with Mild Cognitive Impairment—Designing the MEMO_MOVE PROGRAM

**DOI:** 10.3390/ijerph191610221

**Published:** 2022-08-17

**Authors:** Catarina Alexandra de Melo Rondão, Maria Paula Gonçalves Mota, Dulce Esteves

**Affiliations:** 1Department of Sports, University of Beira Interior, 6201-001 Covilhã, Portugal; 2Fundão City Council, 6230-338 Fundão, Portugal; 3Research Center in Sports Sciences, Health Sciences and Human Development (CIDESD), 5001-801 Vila Real, Portugal; 4Department of Sports, University of Trás os Montes e Alto Douro, 5000-801 Vila Real, Portugal

**Keywords:** exercise program design, multimodal exercise for MCI/dementia, exercise characteristics for MCI/dementia

## Abstract

Dementia patients are at high risk for the decline of both physical and cognitive capacities, resulting in an increased risk of the loss of autonomy. Exercise is regarded as a non-pharmacological therapy for dementia, considering the potential benefits of preventing cognitive decline and improving physical fitness. In this paper, we aim to describe the different design stages for an exercise program combined with cognitive stimulation for a population with mild cognitive impairment, i.e., the MEMO_MOVE program. Methods: The intervention design followed the Medical Research Council’s guidelines for complex interventions and was structured according to the six steps in quality intervention development (6SQuID). The intervention was described considering the Template for Intervention Description and Replication (TIDieR). In order to establish the intervention characteristics, a literature review was conducted to collate and analyze previous work, which provided a summary the type of exercise that should be implemented among this population. Results: The MEMO_MOVE program was structured and described, regarding (i) inclusion of a cognitive stimulation component; (ii) the kind of cognitive stimulation; and (iii) the type of exercise, duration, frequency, intensity, and program length. Conclusions: A systematic step-by-step process design was followed to create a specific intervention to promote physical fitness and cognitive stimulation in individuals with mild dementia.

## 1. Introduction

According to Alzheimer’s Disease International (2021), worldwide, there are more than 10 million new cases of dementia per year, giving rise to a new case every 3.2 s [1].

Nichols and Vos (2021) have estimated that the number of people with dementia will rise from 57.4 million cases in 2019 to 152.8 million in 2050 [2]. Most of this increase will be seen in developing countries; however, 60% of people with dementia live in countries of low and middle income. By 2050, this figure is expected to rise to 71%. The combined prevalence of all causes of dementia, Alzheimer’s disease, and vascular dementia has been reported to be 697, 324, and 116 per 10,000 people, respectively, corresponding to individuals aged 50 years and older. [3].

Dementia is a neurodegenerative syndrome characterized by a decline in functional capacity and cognition [4] consisting of a global and irreversible loss of cognitive ability, accompanied by a reduced ability to perform tasks of daily living as well as a range of further neuropsychiatric symptoms [5]. Dementia progresses at varying rates, affecting individuals in different ways [6]. In the absence of a cure, treatment and care focuses on improving the lives of people with dementia along with their career/family life through early diagnosis, information/counseling, physical and mental health support, and addressing behavioral and psychological problems [7].

Feter et al. (2021) noted that dementia was usually preceded by mild cognitive impairment (MCI), which was an important risk factor for dementia presented as a symptomatic phase of neurodegeneration marked by cognitive impairment [8]. MCI is commonly misunderstood as a normal part of aging and permits a crucial window of opportunity into dementia prevention, planning, and policy [9].

Physical activity (PA), by increasing aerobic capacity and cerebral blood flow, appears to help reduce chronic inflammation in the central nervous system, increasing neuroplasticity and promoting the reorganization of neural circuits [10]. Huang et al. (2022) emphasized that exercise may exert protective effects on cognitive function by (i) increasing levels of growth factors, such as brain-derived neurotrophic factor and insulin-like growth factor 1; (ii) regulating inflammatory cytokines; (iii) alleviating oxidative stress; (iv) increasing cerebral blood flow; (v) reducing Aβ concentration; and (vi) inhibiting tau phosphorylation [11].

High levels of leisure-time physical activity leads to a 64% reduction in the risk of dementia progression in individuals with MCI [8,12], and PA is associated with a lower incidence of MCI and dementia [13,14,15,16]. Thus, PA is considered to be an effective non-pharmacological strategy to mitigate or slow the progression of dementia-related degenerative diseases [17,18,19].

Programs have been developed in recent years to increase cognitive and motor function in older adults with MCI [20,21,22,23,24,25]. These interventions consist of exercise or multimodal programs, including exercise and cognitive stimulation, which are defined as interventions for people with dementia through providing a variety of enjoyable activities as well as an overall stimulus for thinking, concentration, and memory, usually in a social setting such as a small group [26].

The combination of cognitive and physical exercise (simultaneous cognitive-motor dual tasking) appears to have a greater benefit on cognitive and physical functions than one individual (either cognitive or physical) task training exercise in MCI [22,27,28,29,30]. Combining physical training with cognitive training in a functional context may contribute to helping cognitively impaired older adults develop skills for everyday activities, fostering some autonomy [31].

Given the potential benefits of PA in both intellectual skills and functional physical fitness, exercise programs, often combined with cognitive stimulation, have been incorporated into activities proposed for this population and implemented in nursing homes, day centers, community centers, health centers, etc.

Taking into consideration the proliferation of these programs, it is important to consider their objectives as well as what types of exercises and related characteristics should be implemented.

Moore et al. (2016) reinforced this idea, stating that the recent literature presents the benefits of PA interventions for patients with dementia although the amount/type of PA required for these benefits remains unclear [32]. Tait et al. (2017) are of the same mind that regular exercise has shown some cognitive benefits for individuals with MCI and agree that the optimal type and dose (frequency, intensity, duration) of exercise remains unclear [29].

In fact, different types of interventions (exercise type, intensity, duration, free frequency, etc.) lead to different outcomes in terms of functional fitness and cognitive function [15]. Lewis et al. (2020) stressed that in order to prescribe and recommend any exercise intervention, it ws necessary to define its frequency, intensity, type, and time (FITT principle). The amount (ITF) of exercise performed may be a crucial factor for outcomes when considering people with dementia. Due to the diversity of interventions [20], it is important to observe that the relationships of PA interventions in preventing cognitive decline or dementia in older adults are largely insufficient, i.e., different exercises lead to different outcomes.

Therefore, identifying optimal qualitative (namely the type of exercise) and quantitative (e.g., frequency, duration, intensity, and temporal proximity of cognitive and motor training) training characteristics which effectively and sustainably increase individual cognitive reserve and show impact on activities of daily living is an important point to be investigated [33].

In this paper, we aim to describe the process of developing an exercise intervention using formal models of intervention development while establishing a theoretical framework that supports the design of a multicomponent intervention, namely MEMO_MOVE, which is specific, theoretically considered, and evidence-based for people with mild dementia and MCI.

Before implementing the program, the following questions should be addressed:-What kind of exercise intervention and cognitive stimulation seems to be more effective for dementia patients?-What is the FITT plan that best improves physical fitness (functional ability, fitness, walking, equilibrium, muscular strength) and cognitive outcomes (cognitive function and cognitive ability)?-Is there sufficient evidence to favor one or more types of exercise for dementia patients?

This article was constructed in order to help address the fact that there are few published studies on the development of interventions, in other words, it is an article based on originality and novelty.

## 2. Materials and Methods

### 2.1. Intervention Design

To develop an intervention program for a population with MCI, some steps must be taken into consideration. The intervention design followed the Medical Research Council’s guidelines for complex interventions described by Craig et al. (2008) [34], and updated by Skivington et al. (2021) [35]. In order to structure the intervention, the following six steps in quality intervention development (6SQuID) proposed by Wight et al., 2016 were performed: (1) Define and understand the problem and its causes. (2) Identify factors with greatest scope for change. (3) Identify how to bring about change. (4) Identify how to deliver the change. (5) Test and refine on a small scale. (6) Collect sufficient evidence of effectiveness to justify rigorous evaluation/implementation.

With the aim of improving comprehension when reporting the intervention design, in this article, we were guided by the Template for Intervention Description and Replication (TIDieR) checklist and guide [36], considering all 12 items (brief name, why, what (materials), what (procedure), who provided, how, where, when and how much, tailoring, modifications, how well (planned), how well (actual)) [34,35,36,37].

### 2.2. Theorical Framework Development

For the theoretical background of the program, in this article, we followed the work presented by Booth et al. (2018) [38]. Furthermore, a literature review was conducted to collate and analyze previous work investigating exercise interventions in adults with MCI, therefore, providing a clear summary of what types of exercise should be implemented to improve both physical fitness and cognition among this population. We were particularly interested in identifying what FITT plan would best improve physical fitness and cognitive outcomes.

The electronic databases, Web of Science, Scopus, PubMed, and the Cochrane Electronic Library were systematically examined, using the following conjugation of terms: (1) “exercise” or “physical activity”, (2) “mild cognitive impairment” or “dementia”, and “dual task or multimodal”. The search was conducted from September to December 2021.

The results from the literature search were selected from the initial search if they met with the following criteria: (1) included an adult population with MCI or dementia recognized through cognitive testing (e.g., Mini Mental State Examination) or diagnosis (e.g., dementia, Alzheimer’s disease); (2) exercise-based intervention; (3) exercise program completely described (FITT); (4) results concerning physical fitness and cognitive performance; and (5) manuscripts written in English.

Exclusion criteria encompassed: (1) exercise interventions in which the aim was the improvement of other factors other than physical fitness or cognition, such as the improvement of sleep patterns, general health, quality of life, pain management, psychological health, and/or improving psychological parameters, namely mood or depression; (2) lifestyle interventions; (3) revision of exercise guidelines or recommendations; (4) the use of specific exercise programs such as tai chi, yoga, or dance; (5) patients that present psychiatric diseases and other comorbid medical conditions; (6) exercise programs not explicit; and (7) manuscripts written in languages other than English.

Due to the number of articles identified, a further exclusion criterion of articles published before 2015 was introduced to ensure identification of recent evidence and focus on material published after this year.

The process yielded 26 studies to be evaluated (Figure 1), all of which were screened according to the PRISMA systematic review protocol [39,40]. The search was performed by C.A.d.M.R. (first author). Two independent reviewers (D.E. and M.P.G.M.) assessed the included studies for their methodological quality; any discrepancies were discussed until a consensus was definitively reached.

## 3. Results

The rationale for intervention components (”why”) and equipment or procedures used (”what”) were established from the 26 eligible studies. The main result of this article is the intervention description according to the TIDieR checklist.

Name: MEMO_MOVE—Multicomponent exercise intervention for people with MCI.

### 3.1. Why: Rationale, Theory, and Goal (Program Design)

Exercise is a promising nonpharmacological therapy for preventing the decline of cognitive function [11]. Numerous randomized controlled trials [41,42,43] have reported the positive effects of exercise on cognitive function and neuropsychiatric symptoms in patients with cognitive impairment.

Interventional studies have demonstrated that physical exercise improves certain domains of cognitive and affective functions in older adults [24] and potentially enhances brain neuroplasticity [44], preventing cognitive decline. Arrieta et al. (2020) also referred to the fact that physical exercise could enhance hippocampal neurogenesis and neuronal plasticity; thereby, it should counteract the negative effects of aging [45].

In addition to cognitive impairment, older adults often present with physical impairment [22,46], increasing the risk of falls and, in turn, leading to a loss of autonomy. However, the decrease in functional physical aptitude can be delayed by the practice of physical exercise [47,48]. Thus, PA may have a beneficial role in cognitive skills, as well as in the physical fitness of individuals with MCI.

Regarding the type of intervention, a first decision must be considered in our decision tree, i.e., “simple” exercise program or “combined” exercise with cognitive stimulation (Figure 2).

The recent literature has considered that dual task or multicomponent exercise is more advantageous than simple exercise in this population since working simultaneously on the physical and cognitive components allows for more stimuli while enhancing neural regeneration by increasing blood flow to the brain, promoting neural growth, maintaining brain function, and improving brain plasticity [46,49]. A combined physical and cognitive rehabilitation program can lead to significant improvements in physical fitness, also improving cognitive performance [28,31,50,51,52,53,54,55,56,57,58,59,60].

Therefore, our option was to design a multicomponent exercise combined with cognitive stimulation.

Anderson-Hanley et al. (2012) [61] classified combined exercise and cognitive interventions into:“Combined/sequential” interventions, wherein the components are administered sequentially/in tandem [23,28,62,63,64,65];“Dual-task/multicomponent” interventions, wherein components are administered simultaneously, but are typically separate tasks (e.g., reciting numbers backwards while walking) [66,67,68,69,70,71,72,73], which can be further contrasted with;Interactive interventions, as in exergaming, in which the actions in one realm affect the other (e.g., pedaling and steering controls progress in a virtual world and attainment of goals) [61,74,75,76,77,78];According to these authors, a second decision must be considered in our decision tree, i.e., sequential, dual task/multicomponent, or interactive intervention (Figure 2).Tait et al. (2017) concluded that there were inconsistent findings with regard to the cognitive benefits of sequential training as compared with cognitive or exercise training alone. In contrast, simultaneous (dual task) training interventions have significantly improved cognition in both healthy and older clinical populations.

Interactive interventions can provide an integrated physical and cognitive improvement [69,79,80,81]; however, this option cannot be considered due to the reality of the materials available.

Therefore, dual task exercise/multicomponent intervention was the option chosen; cognitive stimulation was incorporated into the sessions, simultaneously combined with resistance or aerobic training.

### 3.2. A Third Decision Concerns the Type of Cognitive Stimulation

Cognitive stimulation is an intervention for people with dementia that offers a very wide variety of enjoyable activities. Providing general stimulation of thinking, concentration, and memory, usually in a social setting such as a small group or even individually, cognitive stimulation involves a range of activities which aim to stimulate cognitive abilities such as attention, memory, language, thinking (including the discussion of past and pre-envisioned events and topics of interest), word games, puzzles, music, and hands-on activities such as indoor baking or gardening [26].

Cognitive training studies have recommended that training for executive functions (e.g., working memory) improved the efficiency of the prefrontal network, which provides support for brain function in the face of cognitive decline. While physical activity preserves neuronal structural integrity and brain volume (hardware), cognitive activity strengthens the functioning and plasticity of neural circuits (software), thus, supporting cognitive reserve in different ways [82].

From the 26 studies retrieved in the literature review, we concluded that the main cognitive stimuli used were: composing poems [46,83,84]; special stair training combined with counting [53,56,84,85]; semantic/phonemic verbal fluency tasks [53,86]; word games [56,83]; mathematical calculations [83,84]; forward and backward counting [85]; memory games (cards, sounds, or movements) [86]; reality-directed orientation training [87]; computer-aided cognitive training with stimulation (oral, memory, verbal fluency, spatial learning, attention, executive functions, and orientation) [28,88]; cognitive exercises based on horticultural intervention [89] and walking while singing and playing instruments [90].

Considering this data in conjunction with the type of equipment available for this intervention, cognitive stimulation comprises exercise based on word games, mathematical calculations, forward and backward counting, computer exercises (Cogweb (www.cogweb.pt, accessed on 10 January 2019) and brain on track (www.brainontrack.com, accessed on 10 January 2019)), exergames (Blaze pod (www.blazepode.com, accessed on 10 January 2019), and balance platforms (Physio Sensing, www.physiosensing.net, accessed on 10 January 2019).

This stimulation is implemented in two sets of exercises: (1) repetitive (we repeated some exercises in order to promote evolution and apply the use of acquired skills) and (2) alternated (in each session, we introduced different cognitive exercises in order to maintain the effect of innovation/surprise and motivation).

For example, one day they performed arithmetic tasks (one or two-digit subtraction) and played a word game (in which the technician said a category and the user had to say as many words related to that topic as possible) while simultaneously performing squats with the help of a chair.

Another day, they tried to change the direction of their walk from front to back or left to right according to a whistle pattern, while counting up or down in twos or threes. In another exercise, they were encouraged to say as many animal names as possible while walking up and down a step. In addition, coordinative circuits combined with cognitive tasks were performed. At one station they had to associate a part of a certain category with a picture, at a second station they performed memory tasks with letters, at a third station they had to say the names of people beginning with a certain letter, and at a fourth station they completed puzzles; along the circuit there were tasks of dynamic balance, static balance, and strength with calisthenics exercises.

### 3.3. The Fourth Decision Addresses the Exercise Characterization (FITT)

Concerning “type”, the intervention within the literature review presented aerobic training, strength training, and functional training as the most commonly used.

For our study, functional training was chosen due to the characteristics of the participants who sometimes present physical weaknesses, making it difficult to apply any aerobic type of training. Our exercise program combined strength, balance, and endurance exercises, as well as cognitive stimulation [22,27,28,31,46,49,52,53,54,56,57,84,91].

Considering “frequency”, according to the literature review, intervention performed from two to three times per week was the usual frequency applied. A frequency of two times per week was chosen for our program as our reference point [56,57,84,85,86,88,92,93].

Regarding “duration (session time)”, programs from Table 1 refer to 45 to 60 min as a regular training program. We adopted the same duration, i.e., 50–60 min [46,51,53,57,59,87].

About “duration (program time)”, the program lasted six months, which is similar to several programs presented in Table 1 [46,51,67,103].

Finally, regarding “intensity”, the literature [53,57,58] recommended light to moderate intensity. In order to individualize the training, a fitness evaluation was initially performed, and the intensity was specifically adapted to each subject. Due to the capacities observed among participants, light to moderate intensity was applied.

### 3.4. What—Materials: Provider, Participant, and Equipment

The MEMO_MOVE provider is a consortium of a university (Universidade da Beira Interior), a research center (Research Center in Sports Sciences, Health Sciences and Human Development), and the Fundão City Council. The consortium aims to partner sport science with a community program tailored to people with MCI/dementia.

Regarding participants, the intervention sessions will be provided in nursing homes. Through the collaboration of the technical directors of the respective institutions, it will be possible to make a preselection of possible participants.

Before being selected for the interventions, all participants (or their families/tutors) must sign an informed consent form.

An initial cognitive evaluation will be performed by a specialized technician in the area of psychology using the Mini Mental State Examination Test Battery [104], which allows for the assessment of the inclusion of the participants in the study as all participants had to have mild cognitive impairment. Regarding physical fitness, the Rikli and Jones test batteries and the Short Performance Battery Test will be applied.

Exclusion criteria include: clinical diagnosis of advanced dementia syndrome, uncontrolled hypertension (BP > 160/90 mmHg), frequent hypoglycemia, severe congestive heart disease, acute myocardial infarction in the past year, severe anemia (HB < 8 g/dL), severe respiratory diseases, severe osteoporosis, sensory deficit (vision/hearing) that prevents collaboration in the physical exercise program, and psychiatric disorders.

Materials and equipment to be used in the MEMO_MOVE program were trialed during a pilot study in which six people with MCI undertook five supervised, multicomponent exercise interventions.

Equipment selected for intervention sessions will be diversified according to the exercises planned for each intervention and will comprise of dumbbells of varying weights, flares, motor skills kits, therapeutic balls, educational games (such as cards, dominoes, and association games), and pedals.

### 3.5. What—Procedure: Provider Training, Assessment, and Intervention Session

The intervention providers include physical education teachers, exercise physiologists, and psychologists. Their training includes theoretical sessions (with sport scientists and neurologists) and practical sessions (a pilot study oriented by senior fitness instructors and exercise physiologists).

Regarding assessment, participants are divided between an experimental group (EG) and a control group (CG).

Cognitive evaluation is performed using the Mini Mental State Examination Test Battery and physical fitness is evaluated by the Rikli & Jones Test Battery [104] and Short Physical Battery Test [105], applied by a physical education teacher and PhD student.

Blood samples are collected in collaboration with a clinical laboratory (Affidea, www.affidea.pt (accessed on 15 January 2019)).

Assessments are made before (baseline) and after intervention.

Participants’ presence in the session and the exercises performed is recorded daily.

The intervention focuses on the application of exercises that aim to work some physical and cognitive capacities within the dual task model, with the physical capacity as the main task and the cognitive capacity as the secondary one.

Functional exercises are performed within functional circuits: upper and lower limb strength using dumbbells or the weight of the body (calisthenics); dynamic and static balance and flexibility; some cognitive functions such as attention, memory, calculation, language, and executive functions were worked on as well.

### 3.6. Where Intervention Location

The intervention sessions will occur in nursing homes ((1) Centro Paroquial e Bem-Estar Social de Valverde, (2) Hotel Senior Prestige do Fundão, and (3) Centro Paroquial de Assistência de Souto da Casa).

### 3.7. How: Method of Delivery

The intervention is delivered face-to-face in a group setting with exercises individualized to the users’ conditions.

To increase motivation and session participation, the self-determination method is used [38] which involves:-Communication strategies, for instance, taking the time to speak and hear the participant; using clear, simple language to explain the exercises; acknowledging the participant’s feelings and mood; using verbal and body language, using a friendly secure voice; promoting participant’s communication.-Motivational strategies, for example, recognizing progress and skills, providing positive feedback, adapting exercise complexity to participants’ skills, using colorful materials, using music and images related to the cultural/social backgrounds of participants, providing different exercises, stimulating socialization in small groups.

### 3.8. Tailoring the Intervention

According to Booth et al. (2018), a central principle of the intervention is that it is individually tailored to the abilities, comorbidities, interests, and goals of the participant. The authors emphasized the need to adapt interventions to overcome content-specificity barriers [38].

The MEMO_MOVE program was tested in a feasibility study, which will be the basis for further refinement in the future.

Some barriers encountered in the development and implementation of the program include:-The motivation of individuals with MCI to exercise and perform the cognitive tasks;-The resistance of family members to participation in the program;-Difficulty in the medical diagnosis of dementia;-Logistical difficulties in participating in the program (transport, timetables).

How these barriers were addressed:-Promotion of a one-to-one trial session, where an affective/trusting bond and proximity with the individual was created. The interpersonal relationship has proven to be a fundamental factor for adherence and participation in the program. The exercises should have a playful character and fit in with the cultural and social experiences of each person.-Inviting families to participate in the sessions, explaining the advantages in terms of functional autonomy and brain stimulation.-Integration of a team of psychologists who help to improve the diagnosis of individual’s cognitive capacities, as well as help to adapt cognitive stimulation to each person.-Choosing a central and easily accessible location, with timetables that are compatible with the lifestyles of the participants.

## 4. Discussion

### 4.1. Summary of Findings

A dual task training intervention program combining physical exercise and cognitive stimulation was developed to support people with MCI, aiming to help participants to improve both physical and cognitive abilities. A literature review was used to design the program. It is suggested that the intervention should include some application criteria as follows: a sample base ranging from 30 to 100 participants; a program duration of 24 weeks; a frequency of twice a week; and the intensity should vary between light, moderate and intense while always taking into consideration the physical frailties of the users.

Based on this analysis, the MEMO_MOVE intervention program was designed to be delivered to populations living in communities with MCI or in senior housing and subsequently be self-managed.

### 4.2. Strengths and Limitations

As previously mentioned, the MEMO_MOVE program resulted from a consortium between the University of Beira Interior, CIDESD, and Fundão City Council. This is an advantageous point as it allies scientific research with the ability to intervene in municipal community programs.

Another strong point is the people responsible for the program, including scientific leaders (sport scientist, PhD), the people responsible for the orientation and implementation (PhD student), and the consultants (psychologists and technicians); the capacity of the multidisciplinary team that designed and implemented the program is fundamental to its success.

A pretest was conducted to evaluate the materials used and the type of cognitive stimulation with more acceptance. This was very important because it showed the motivation, communication skills, and relationship of the intervention team with this population.

As for the limitations, the main one to mention has to do with the lack of a more specific diagnosis of dementia, since the population recruited was identified either by a family doctor or by psychologists from the nursing homes without a defined medical diagnosis of the type and state of dementia.

Another limitation has to do with the acceptance of the program by both family members and nursing home managers.

## 5. Conclusions

### Implementation and Recommendations

As referenced by Booth et al. (2018) [38], the Council’s complex intervention guidelines are broad, providing a general framework that covers all aspects of development from theory identification to evaluation by randomized controlled trial and implementation. However, there is little detail, especially in the critical early stages. In general, there are few published intervention development studies [106].

This article was constructed in order to contribute to closing this gap in the literature.

The developed intervention is not yet sufficiently researched to be recommended as a clinical exercise program, as its effectiveness has not yet been evaluated and the results need to be investigated.

In any case, the development process within the intervention has reached its goals and the described intervention is ready to be implemented, starting as a feasibility trial through a pilot study preferably with the participation of 20 to 30 individuals.

As further evaluation of the effectiveness of the program’s implementation is required, investigation of any obtained future results is recommended.

## Figures and Tables

**Figure 1 ijerph-19-10221-f001:**
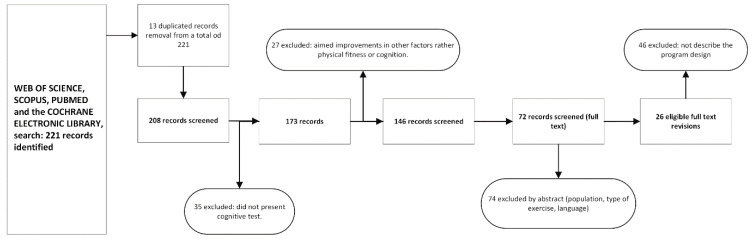
Flow diagram of the review articles.

**Figure 2 ijerph-19-10221-f002:**
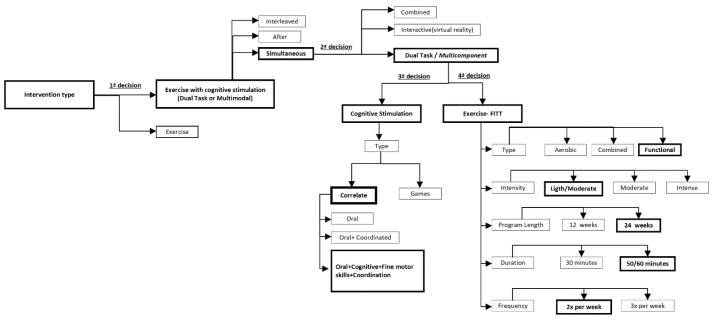
MEMO_MOVE: design decision tree.

**Table 1 ijerph-19-10221-t001:** Literature review: summary of findings.

No	Study ID	Sample Size	Diagnosis/Test	Duration (Weeks)	Frequency	Intensity	Exercise Component	Cognitive Component	Cognitive/Motor Outcome	Control Group
1	Makizako et al., 2012 [94]	50	Dementia/MMSE	24	90 min/2 days a week	Moderate	Aerobic exercises, strength training, balance retraining	Poem composition, stairs stepping while counting 3 backwards, waking on balance board while counting 3 backward	Gait functions	Standard care
2	Suzuki et al., 2012 [83]	50	MCI/MMSE	52	90 min/3 days a week	Moderate	Aerobic exercises, strength training, balance retraining	Poem composition, special ladder training	Gait functions, memory, executive function	Health education
3	Coelho et al., 2013 [67]	27	Alzheimer/MMSE	16	60 min/3 days a week	65% to 75% HRmax moderate	Strength/resistance training, aerobic capacity, flexibility, balance, agility	Cognitive activities focused attention, planned organization, abstraction, motor sequencing, and mental flexibility	Frontal cognitive function	Standard care
4	Suzuki et al., 2013 [95]	100	MCI//MMSE	24	90 min/2 days a week	Not clear	Aerobic exercises, balance retraining, dual task training	Poem creation, special ladder training	Gait functions, memory	Health education
5	Gill et al., 2016 [53]	44	MCI/MoCA	26 plus/26 follow up	50 + 45 min/2 or 3 days a week	Not clear	Aerobic exercises, lower extremity strength training	Semantic/phonemic verbal fluency tasks, random athematic calculations	Gait functions, memory, executive function	Standard care
6	Heath et al., 2016 [96]	63	Dementia/MMSE	24	60 min/3 days a week	Moderate to high intensity (65–85% HRmax)	Aerobic exercise, strength/balance training	Special square stepping exercise involving memory, executive functions, pattern recognition	Executive functions	Standard care
7	Öhman et al., 2016 [85]	210	AD/MMSE	52	60 min/7 days a week	Not clear	Aerobic, strength, endurance, and balance training	Dual tasking with, ball games, dancing, calculation, and memory games	Gait functions, executive function, memory	Standard care
8	Venturelli et al., 2016 [90]	80	AD/MMSE	12	60 min/5 days a week	Not clear	Brisk walking	Reality orientation method	Gait functions	Standard care
9	Silva et al., 2017 [97]	127	AD/MMSE	24/28 follow up	60 min/3 days a week	Moderate to high intensity (65–85% HRmax)	Aerobic exercises, resistance training, stretching	Mind motor training (special square stepping games)	Gait functions	Standard care
10	Delbroek et al., 2017 [89]	20	MCI/MoCA	6	30 min/2 days a week	Not Clear	BioRescue training, aerobic exercises, balance retraining, weight bearing	Memory games, attention maze, dual task training	Gait functions, gait and balance	Standard care
11	Gregory et al., 2017 [98]	56	Dementia/MMSE	26	40 min/3 days/week	Moderate to high intensity (65–85% HRmax)	Aerobic exercise	Executive function training: verbal fluency, memory, and arithmetic	Cognitive functions and gait	Standard care
12	Hagovská et al., 2017 [99]	80	MCI/ACE	10	60 min/2 days a week	Not clear	Leg strengthening exercises, balance training	CogniPlus, memory, attention, executive function, visual motor training	Gait functions, memory, executive function, attention, balance	Health education
13	Karssemeijer et al., 2017 [28]	742	MCI/MMSE	12	30–120 min/3 days a week	Not clear	Aerobic training a single	Computer-aided; cognitive (oral, memory, verbal fluency, spatial learning, attention, executive functions, orientation	Cognitive and motor functions	Standard care
14	Maffei et al., 2017 [100]	113	MCI/MMSE	28	60 min/5 days a week	Not clear	Aerobic exercises, strength, control, and flexibility training	Attention, memory, learning, and meta cognitive training	Gait functions, executive function, attention	Non-musical cognitive task and walk
15	Shimada et al., 2017 [92]	308	MCI/RAVLT/MMSE	40 plus/+ 12 follow up	90 min/1 day a week	Not clear	Aerobic exercises, balance retraining, dual task training	Cognitive training (horticultural intervention)	Gait functions, memory, executive function	Standard care
16	Chen et al., 2018 [93]	28	Dementia/MMSE	8	60 min/1 day week	Not clear	Functional activities, dual task walking	Walking while singing, playing instruments, cognitive load stepping	Executive function, balance, gait functions,	Standard care
17	Donnezan et al., 2018 [91]	69	MCI/MMSE	12	60 min/2 days week	Moderate 60% HRmax	Aerobic training on bicycles, balance retraining.	Game software “HAPPYneuron” and Presco	Attention, executive functions, balance, gait functions	Standard care
18	Wiloth et al., 2018 [101]	99	Dementia/MMSE	10	90 min/2 days week	Not clear	Dual task walking, sit to stand maneuver	Game based training (motor-cognitive exercises)	Executive functions	Standard care
19	Lemke et al., 2019 [88]	105	Dementia/MMSE	10 plus/+12 follow up	90 min/2 days week	Not clear	10 m walk, dual tasking	DT Serial low/high demand calculation (2–3 forward and backward calculation)	Gait functions	Standard care
20	de Oliveira Silva at al., 2019 [102]	52	MCI/AD/CDR	12	60 min/2 days week	70–80% (VO_2max_) or 80% of HRmax	Balance, aerobic, and strength training and stretching	Executive functions, verbal training, selective attention	Mobility and executive function	Health education
21	Park et al., 2019 [56]	49	MCI/MMSE	12	110 min/2 days week	Moderate	Aerobic exercises included stair stepping, resistance walking and stair climbing, and agility stair walking	Word games and numerical calculations	Cognitive function and physical function	Standard care
22	Rezola-Pardo et al., 2019 [87]	85	MCI/AD/MMSE	12	60 min/2 days week	Moderate	Strength and balance exercises	Verbal training and arithmetic calculation	Physical performance and gait speed, cognitive functions	Standard care
23	Zhang et al., 2019 [31]	Not clear	MCI/	Not clear	Not clear	Not clear	Strength and balance training	Attention and executive function	Cognitive and motor functions	Not clear
24	Bae et al., 2020 [84]	280	MCI/MMSE	40	90 min/1 day week	Moderate	Aerobic exercises, balance strength training	Calculation, word games, poems citing, challenging cognitive tasks	Gait Functions, memory, executive function, motor functions	Standard care
25	Parvin et al., 2020 [57]	32	DA/MoCA	12	40 to 60 min/2 days week	Moderate	Muscle endurance, balance, flexibility, and aerobic exercises	Short-term and working memory, attention and executive function	Cognitive and motor function	Standard care
26	Kim et al., 2021 [86]	20	MCI/MMSE	12	60–90 min/1 day week	Moderate	Strength, rubber band	Remembered the names and main uses; counting numbers; planning and solving complex story problems	Balance and gait	Standard care

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
