# Peer review of "Development of a Combined Exercise and Cognitive Stimulation Intervention for People with Mild Cognitive Impairment—Designing the MEMO_MOVE PROGRAM"

_ijerph, 2022, doi:10.3390/ijerph191610221_

Round 1
Reviewer 1 Report
The article deals with the development of a stimulation program that combines physical activity with cognitive stimulation among patients with mild cognitive impairment. The authors may consider the following comments:
The paper presents the assumptions about the MEMO MOVE program. In the work, we may find a literature analysis on similar interventions. One of the basic drawbacks of the work is the lack of a clear definition of what parameters the program will be characterized by - that is, what interventions will be implemented in the respondents. In addition, the decision tree is quite unclear; it is difficult to determine on what basis the respondent will go through the various stages of decisions regarding the implemented procedures.
One of the inclusion criteria for the study was the presence of mild cognitive impairment assessed by MMSE. Are there similar inclusion / exclusion criteria for physical activity? The study concerns the elderly, so will it take into account comorbidity, e.g. frailty or sarcopenia as exclusion criterion?
Minor comments: The manuscript should be reviewed by a native speaker. The text contains typos, inappropriate capital letters within sentences, and, in some places, a period/full stop is missing at the end of the sentence.
Author Response
I attach the changes suggested by the reviewer.

Reviewer 2 Report
Thank you for the opportunity to review this interesting article by Rondão C. and colleagues.
The topic is interesting and relevant to the actual ageing world. The article is interesting and the authors clearly explaining every aspect of the research. The methodological approach is described and the structure of the paper has a logical flow. The findings are properly addressed.
I only have some concerns/improvement suggestions related to the paper:
1. As mentioned in chapter 4.3, the lines of the Medical Research Council guide for complex interventions ”are broad, providing a general framework design,… there are few published intervention development studies… This article was constructed in order to contribute to closing this gap in the literature”. On these aspects are based the originality and novelty of the paper and this would be good to be underlined in the Introduction;
2. 2. The content of the chapters of the article would be also welcome;
3. 3. “Table 1. Literature review: summary of findings” is referred twice in page 6 and included only in page 8. For a better understanding of the results of the paper, it would be useful to add a Supplementary material chapter at the end of the paper and include Table 1 in it;
4. 4. Recommendations for further research are also of interest. That is why, in chapter 4.3, it would be recommended to underline the necessity of the evaluation of the effectiveness of the approach and of the investigation of the results as a further work.
5. 5. I would like to point out two small writing errors: (a) in lines 154-155 “Due to the number of articles identified, a further exclusion criterion of articles published before 2015 was introduced to ensure identification of recent evidence and capture of material published prior to those dates” , I think that “prior to those dates” must be replaced with “after this date”; (b) in line 254 “Tis stimulation is implemented in two sets of exercises”, I think that “This” would be used instead of “Tis”.
I hope my feedback is useful to the authors in improving their paper and wish them all the best in pursuing this important area of research.
Author Response

(The authors gave the same response as above.)

Round 2
Reviewer 1 Report
Comment 1. What the text from the Response to Reviewers Comments refers to?
“Response to Reviewers Comments Manuscript ID: 950937
Regarding our submission in Frontiers in Aging Neuroscience, section Alzheimer's Disease and Related Dementias entitled “Multicomponent exercise program effects on fitness and cognitive function of elderlies with mild cognitive impairment: involvement of oxidative stress and BDNF” (Manuscript ID: 950937), in order to fulfil the editors’ request to reply to reviewers, authors are submitting a detailed response to each of the comments and suggestions, explaining how we addressed each concern in our revised submission.”
Comment 2. Some sentences after amendments have lost their consistency.
Line 33 Nichols and Vos (2021) estimate that the number of people with dementia number predicted from 57.4 million cases in 2019 to 152.8 million in 2050 [2]
Line 73 Combining physical training with cognitive training in a functional context may contribute to helping cognitively impaired older adults develop training to everyday activity, creating some autonomy[31]
Comment 3. One-sentenced paragraph
Lines 232-237 Cognitive stimulation is an intervention for people with dementia that offers a very wide variety of enjoyable activities that provide general stimulation of thinking, concentration, and memory, usually in a social setting, such as a small group, or even individual, involving a range of activities that aim to stimulate cognitive abilities such as attention, memory, language, thinking, including discussion of past and pre-envisioned events and topics of interest, word games, puzzles, music, and hands-on activities such as indoor baking or gardening [26].
Comment 4: Line 255 “will comprehend exercised based on word games” do you mean “exercises”
Comment 5. Line 307 the sentence” For physical fitness all elderly with autonomy and mobility were considered” was added as the response to my question. Still it hasn't answered my question. I want to know if the physical fitness was determined on the basis of any specific tool, scale, questionnaire. You use the term autonomy and mobility, but this is a term that applies to both people who are completely physically fit and those who move with a cane or walking frame. Yet there is a difference between them.
Comment 6
Crig et al., (2008) Wight et al, 2016- Arrieta et al. (2020) Tait et al (2017)
I would like to know what is the correct way to write this abbreviation.
Summary: First, if the text is not checked by a native speaker, it is not suitable for publication. Secondly, the responses for the reviewer should refer to the reviewed work. And they should contain any content. If the author does not agree with the reviewer's suggestion, he should articulate it. However, writing the sentence "Correction made as suggested by the reviewer" for which there is no answer to the question asked neither in the text nor in the "responses to the reviewer" gives a bad impression about the revision of the article.
Author Response
You can find the new version, with the corrections in the attachement.
Comment 1. What the text from the Response to Reviewers Comments refers to? “Response to Reviewers Comments Manuscript ID: 950937 Regarding our submission in Frontiers in Aging Neuroscience, section Alzheimer's Disease and Related Dementias entitled “Multicomponent exercise program effects on fitness and cognitive function of elderlies with mild cognitive impairment: involvement of oxidative stress and BDNF” (Manuscript ID: 950937), in order to fulfil the editors’ request to reply to reviewers, authors are submitting a detailed response to each of the comments and suggestions, explaining how we addressed each concern in our revised submission.”
Comment 2. Some sentences after amendments have lost their consistency. Line 33 Nichols and Vos (2021) estimate that the number of people with dementia number predicted from 57.4 million cases in 2019 to 152.8 million in 2050 [2] Line 73 Combining physical training with cognitive training in a functional context may contribute to helping cognitively impaired older adults develop training to everyday activity, creating some autonomy[31]
Comment 3. One-sentenced paragraph Lines 232-237 Cognitive stimulation is an intervention for people with dementia that offers a very wide variety of enjoyable activities that provide general stimulation of thinking, concentration, and memory, usually in a social setting, such as a small group, or even individual, involving a range of activities that aim to stimulate cognitive abilities such as attention, memory, language, thinking, including discussion of past and pre-envisioned events and topics of interest, word games, puzzles, music, and hands-on activities such as indoor baking or gardening [26].
Comment 4: Line 255 “will comprehend exercised based on word games” do you mean “exercises”
Comment 5. Line 307 the sentence” For physical fitness all elderly with autonomy and mobility were considered” was added as the response to my question. Still it hasn't answered my question. I want to know if the physical fitness was determined on the basis of any specific tool, scale, questionnaire. You use the term autonomy and mobility, but this is a term that applies to both people who are completely physically fit and those who move with a cane or walking frame. Yet there is a difference between them. Comment 6 Crig et al., (2008) Wight et al, 2016- Arrieta et al. (2020) Tait et al (2017) I would like to know what is the correct way to write this abbreviation. Summary: First, if the text is not checked by a native speaker, it is not suitable for publication. Secondly, the responses for the reviewer should refer to the reviewed work. And they should contain any content. If the author does not agree with the reviewer's suggestion, he should articulate it. However, writing the sentence "Correction made as suggested by the reviewer" for which there is no answer to the question asked neither in the text nor in the "responses to the reviewer" gives a bad impression about the revision of the article.
|
- Regarding our submission in ijerph entitled “Development of a combined exercise and cognitive stimulation intervention for people with mild cognitive impairment – Designing MEMO_MOVE PROGRAM” (Manuscript ID: ijerph-1826002), in order to fulfil the editors’ request to reply to reviewers, authors are submitting a detailed response to each of the comments and suggestions, explaining how we addressed each concern in our revised submission.
Comment 2- Revised by a native speaker;
“Nichols and Vos (2021) estimate that the number of people with dementia will rise from 57.4 million cases in 2019 to 152.8 million in 2050” “Combining physical training with cognitive training in a functional context may contribute to helping cognitively impaired older adults develop skills for everyday activities, fostering some autonomy[31].” Comment 3- Revised by a native speaker; “Cognitive stimulation is an intervention for people with dementia that offers a very wide variety of enjoyable activities. Providing general stimulation of thinking, concentra-tion and memory, usually in a social setting such as a small group, or even individually, cognitive stimulation involves a range of activities which aim to stimulate cognitive abili-ties such as attention, memory, language, thinking (including the discussion of past and pre-envisioned events and topics of interest), word games, puzzles, music, and hands-on activities such as indoor baking or gardening [26].”
Comment 4: - No, this is a board game. The game consists of creating words using isolated letters. Comment 5- To evaluate physical fitness, we use the Rikli and Jones and Short Performance Battery Test batteries.
Comment 6 The following abbreviation was considered: Craig et al., (2008) Wight et al., 2016- Arrieta et al., (2020) Tait et al., (2017) - The text was revised by a native speaker, and some changes were made to improve the sentences |
